# Relative importance of socioecological domains to predicting opioid-involved mortality

**Joshua C. Black** *, **Annika M. Czizik**

Rocky Mountain Poison & Drug Safety, Denver Health and Hospital Authority, Denver, Colorado, United States of America

* joshua.black@rmpds.org

## Abstract

### Background

The opioid crisis in the United States is a complex issue with interconnected factors that lead to opioid misuse and opioid-involved mortality. This study assessed the relative importance of different risk factor domains in predicting fatal opioid-involved mortality that occurred after hospital encounters involving opioids.

### Methods

A machine learning model was developed by integrating multiple data sources, including hospital records, death records, and societal data. The model allowed simultaneous examination of risk factors across individual drug and non-drug related factors, hospital factors, and societal factors.

### Results

429,005 patients with opioid-related encounters in 2014 were assessed, where 56.6% were female and the mean age was 44.98. Among deaths that had specific drugs listed for both the hospital encounter and the death, 51.7% of hospital encounters progressed to a more potent opioid at death. Community factors cumulatively had similar importance as individual drug-related factors in predicting opioid-involved deaths and were relatively more important in predicting opioid-involved mortality compared to non-drug involved mortality. In predicting opioid-involved mortality, non-drug related individual-level predictors accounted for 45.1% of the importance. Community factors accounted for 27.9% of the importance and drug-related individual factors accounted for 22.5%. In contrast, community factors accounted for only 16.5% of the importance when predicting non-opioid-involved mortality.

**Data availability statement:** Restricted data are available through the CDC Research Data Center (RDC), at https://data.cdc.gov/NCHS/Drug-Involved-Mortality-DIM-data-available-through/px5t-5gtx/about_data. Access can be obtained through submission of proposals to the RDC, at https://www.cdc.gov/rdc/application-process/index.html.

**Funding:** JCB was supported by the Centers for Disease Control and Prevention through an RFP titled: Call for Proposals for Using Linked Hospital and Mortality Data to address the Opioid Crisis (https://www.cdc.gov/nchs/data/nhcs/FY18-PCORTF-Call-for-Proposals-508.pdf). The CDC played no role in the study design, analysis, nor decision to publish. The manuscript was reviewed by representatives from the CDC to verify no disclosure of personally identifying information occurred, but otherwise did not participate in preparation of the manuscript. JCB and AC were employed by Denver Health and Hospital Authority (DHHA), an academic safety net hospital and a nonprofit, political subdivision of the US State of Colorado. DHHA provided support in the form of salaries for authors, but did not have any additional role in the study design, data collection and analysis, decision to publish, or preparation of the manuscript. The specific roles of these authors are articulated in the 'author contributions' section. There was no additional external funding received for this study.

**Competing interests:** This research was conducted by JCB and AC, who were employed by Rocky Mountain Poison and Drug Safety (RMPDS), a department of nonprofit Denver Health and Hospital Authority (DHHA), a subdivision of the US State of Colorado. Outside of this work, RMPDS is supported by subscriptions from pharmaceutical manufacturers, government, and non-government agencies for surveillance, research and reporting services. No subscriber participated in the conception, analysis, drafting, or review of this manuscript. This does not alter our adherence to PLOS ONE policies on sharing data and materials.

## Practice Implications

Rather than suggesting community factors outweigh individual factors, our results highlight individual vulnerability may be amplified or mitigated by broader environmental factors. Interventions targeting larger social determinants of health may be strongly influential in reducing drug-involved mortality. This study demonstrated a quantitative evaluation of the different domains of risk factors and highlighted the importance of considering societal and community factors in a holistic approach to preventing opioid-involved mortality.

## Introduction

The opioid crisis in the United States (US) is a complex issue characterized by interconnected factors that influence patterns of patients using medications for pain, misuse for nontherapeutic reasons, and an alarming number of annual fatalities involving opioids. Previous studies have identified individual risk factors for opioid-involved mortality, such as past substance abuse, psychiatric disorders, and increased healthcare utilization as predictors of overdose [1–4]. Community factors such as reduced market supply, increased accessibility of naloxone, and implementation of fentanyl testing have been associated with a decrease in overdose incidence [5]. Large-scale social factors of community economic distress, rates of incarceration, treatment availability, and cannabis accessibility measures are also predictive of overdose [6]. A social-ecological framework for the opioid overdose crisis has summarized these multiple domains that influence opioid use and mortality, where domains are nested within each other [7]. A quantitative, systems-level understanding of the relative importance of these domains is needed to create a comprehensive picture of trajectories toward opioid-involved mortality [8].

Machine learning techniques have been used to quantify relationships between large sets of factors. Prior machine learning studies investigating overdose predictors have largely focused on individual patient characteristics, omitting broader socio-environmental determinants [9–11]. Since community and societal factors were not included, these studies were unable to quantitatively evaluate the relative contribution across different domains of risk factors.

To address this gap, we developed a machine learning model that integrates individual, hospital, and community-level factors to predict opioid-involved mortality among patients with hospital encounters involving opioids. Unlike previous studies, this enables a comparative assessment of how different risk domains contribute to mortality prediction. This is critical for guiding targeted intervention strategies and determining whether community factors contribute meaningfully to reducing opioid-involved mortality compared to individual drug-related risk factors [1]. The objective of this study was to quantify the relative importance of individual, hospital, and community factors in predicting progression from a hospital encounter involving opioids to opioid-involved mortality. Additionally, we aimed to provide a policy-relevant framework for addressing opioid mortality by identifying risk factors at multiple different

intervention levels. Importance was compared between predicting non-drug involved mortality, opioid-involved mortality, and progression from low potency opioids at hospital encounters to higher potency opioids involved in death.

## Methods

### Data sources and management

A total of 5 different data sources were integrated for the analysis in this study. A restricted dataset was previously created by the National Center for Health Statistics (NCHS) by linking together patient-level data from the first three data sources: the National Hospital Care Survey (NHCS), the National Death Index (NDI) and Drug-Involved Mortality (DIM) datasets (herein, the NCHS-NDI-DIM dataset) [12]. As a secondary data analysis of anonymized data, review by institutional boards was not required. The proposal was reviewed by the Centers for Disease Control and Prevention (CDC) research data center.

The NHCS collects data from inpatient discharges and in-person visits at hospitals. Hospitals were sampled to cover a variety of bed sizes, types, and rurality, although it is not a nationally representative sample, as described in previous literature [13]. The 2016 NHCS was used to identify a cohort of opioid-involved hospital encounters. Opioid involvement was determined using International Classification of Diseases, 10th Revision, Clinical Modification (ICD-10-CM) diagnosis codes. Individual opioids involved at the hospital encounter were identified, and potency was ranked based on equianalgesic equivalence factors (S1 Table 1) [14]. The NCHS-NDI-DIM dataset with geographic information is a restricted dataset, and required access via a research data center.

The NDI is a comprehensive database of death records in the US, containing information about cause of death [15]. Linkages from the NHCS to the NDI identified individuals who died in 2016–2017 (primary outcome) during or subsequent to the hospital encounter; lack of a linkage indicated no death had occurred. The DIM supplements the NDI. It consists of literal text terms extracted from cause-of-death fields on death certificates, where drugs listed in the cause-of-death fields were assumed to be causally involved in the death [16]. It was used to identify specific opioids involved in the death and to stratify the death outcome by whether it was opioid-involved and if the individual progressed from a lower potency opioid at the hospital encounter to a higher potency opioid at death. Opioid involvement in the DIM was broadly defined and included overdose (i.e., poisoning), adverse reactions, misuse, unspecified exposures, and more [17].

Two supplementary datasets were used to quantify societal factors. The Robert Wood Johnson Foundation (RWJF) aggregates county-level health-related and social environmental data from across multiple public data streams, such as data related to health behaviors, access to care, quality of life, social resources, and economic factors [18]. Additionally, data on prescriptions dispensed from retail pharmacies, which quantified total prescription opioids available in a local area by active ingredient, were obtained from IQVIA's US-Based Longitudinal Prescription Data (Danbury, CT).

For analysis, all five data systems must be integrated at the same geographic level. The NCHS-NDI-DIM and IQVIA datasets were obtained at the 3-digit ZIP code level. To link RWJF county data to ZIP codes, a weighted average for each variable was calculated using resident overlap files from Housing and Urban Development [19]. For each ZIP code, the proportion of the population that overlapped with each contributing county was calculated, such that the sum of weights per ZIP code was unity. The weighted average for each RWJF variable in each ZIP code is therefore interpreted as the population-weighted average of the counties comprising the ZIP code. The ZIP-level RWJF weighted averages and IQVIA datasets were then merged onto the NCHS-NDI-DIM dataset at the ZIP code level.

### Statistical analysis

S1 Table 2 summarizes how factors were organized into domains for analysis. Drawing from previous socioecological models as guides [6,7], the variables were categorized into four groupings based on the type of effect and socioecological level. These frameworks were selected because they conceptually represent different levels where policy intervention may

be targeted. Variables available in the datasets were organized into groups similar to those described in the frameworks. The first grouping combined individual demographics and non-drug related ICD codes, including non-drug related ICD-9-CM codes (e.g., acute myocardial infarction, I21), type of admission (e.g., inpatient vs. emergency), payment type, days of care, binary sex, and age. The second grouping contained individual drug-related factors, which include drug-specific ICD codes (e.g., poisoning by heroin, T40.1) and individual drugs identified with the hospital encounters [12]. The third grouping consisted of hospital-level factors (e.g., number of beds indicator, rurality indicator). Finally, community factors comprised a fourth group, including proxies for built environment (e.g., food insecurity) and affluence (e.g., median income), health metrics (e.g., providers per population), and total prescription drug dispensing from the IQVIA Longitudinal Patient Database.

A random forest (RF) machine learning model was developed to calculate the importance of each factor grouping in predicting mortality. Because RF is a nonparametric, ensemble decision-tree model, it is well suited to integrate predictive factors from multiple levels, which likely do not all have the same parametric relationship with mortality. While the RF model cannot quantitate the association effect size between individual variables and mortality, the relative importance of each factor to predicting mortality can be calculated by estimating the GINI impurity for each variable, which has been used to examine overdose predictors in adolescents [10]. The importances of individual factors were summed within the four groups. A predictive RF model was constructed for each of three binary outcomes: non-drug involved death, drug-involved death, and deaths where individuals progressed from a lower potency drug at the hospital encounter to a higher potency drug at death (among drug-involved deaths only). Non-drug involved death was modelled as a comparator for evaluating domain importance against all-cause mortality, excluding drug-involved mortality. Out-of-bag prediction accuracy is reported for each model, with 40% held out-of-bag per tree. Throughout, summary data are suppressed if counts were less than 10, consistent with NCHS policy. Statistics were calculated using SAS 7.4.

**Table 1. Patient demographics, encounter counts and opioids present in hospital encounters.**

| Opioids | All Opioid Encounters | | | | Specific Opioids Identified for Both Hospital and Death | |
|---|---|---|---|---|---|---|
| | Male | Female | Age Mean (SD) | Number of Linked Deaths | Number of Encounters | Deaths with Higher Potency Opioid Involved N (%) |
| All opioids | 185,805 | 242,849 | 44.98 (20.40) | 2,129 | 1,144 | 591 (51.7) |
| Buprenorphine | 559 | 783 | 36.70 (13.27) | 51 | N/Aᵃ | N/Aᵃ |
| Codeine | 6,729 | 12,832 | 45.52 (20.99) | 30 | 20 | Suppressed |
| Fentanyl | 34,823 | 37,145 | 43.40 (23.14) | 976 | 200 | --ᵇ |
| Heroin | 3,809 | 2,027 | 35.24 (12.14) | 881 | 270 | 169 (62.6) |
| Hydrocodone | 18,826 | 28,186 | 46.15 (19.22) | 115 | 72 | 61 (84.7) |
| Hydromorphone | 48,716 | 61,616 | 46.83 (17.25) | 46 | 379 | 151 (39.8) |
| Levorphanol | 533 | 667 | 19.34 (18.96) | Suppressed | 0 | -- |
| Meperidine | 1,127 | 2,031 | 47.35 (20.41) | Suppressed | 0 | -- |
| Methadone | 3,449 | 3,805 | 35.60 (18.05) | 148 | 63 | 21 (33.3) |
| Morphine | 68,965 | 102,610 | 43.92 (20.80) | 193 | 415 | 371 (89.4) |
| Oxycodone | 6,320 | 9,088 | 48.03 (18.70) | 299 | 53 | 37 (69.8) |
| Oxymorphone | 46 | 59 | 51.93 (14.43) | 63 | 0 | -- |
| Tramadol | 8,384 | 13,801 | 48.81 (19.91) | 54 | 42 | 36 (85.7) |

ᵃBuprenorphine not analyzed because relative potency is not well defined.

ᵇFentanyl was the most potent drug, and therefore hospital encounters could not progress to a more potent opioid.

## Results

A total of 429,005 patients with opioid-related encounters were analyzed, where 56.6% (n = 242,849) were female and the mean age was 44.98 (Table 1). The top three opioids encountered were morphine, hydromorphone, and fentanyl. Among deaths that had specific drugs listed for both the hospital encounter and the death, 51.7% (n = 591) of hospital encounters progressed to a more potent opioid at death. A total of 103 hospitals were included in the sample (Table 2). Diverse hospital characteristics were included, with a mix of rurality, service type, size, ownership, and geographic regions.

Out-of-bag prediction accuracy was high for predicting non-drug involved deaths (94.3%) and opioid-involved deaths (99.3%). Accuracy was lower for the model predicting progression to more potent opioids (80.9%). Table 3 describes the highest importance factors in predicting mortality with percentage of importance explained by the variable. In the non-drug involved mortality model, the most important individual factor was neoplastic-related diagnostic codes; the most important community factor was the level of drinking. Neoplastic diagnostic codes (Importance = 22.0%), age (17.9%), and cardiovascular diagnostic codes (4.0%) accounted for 43.9% of the importance in model prediction of non-drug-involved mortality. Diagnostic codes were less important in predicting drug related death compared to their importance predicting non-drug involved mortality among individual non-drug related factors.

When grouped, the importances of community factors and individual drug-related factors were higher when predicting opioid-involved mortality than non-drug involved mortality (Fig 1). When predicting opioid-involved mortality, non-drug related predictors were most important (45.1%), with age being a major factor (7.3%). Community factors accounted for 27.9% of importance, while individual drug related factors accounted for 22.5%. In contrast, when predicting non-drug involved mortality, individual non-drug factors accounted for 76.0% of the importance of the model and community factors

**Table 2. Hospital characteristics.**

| Characteristic | All Hospitals (N) |
|---|---|
| Total (N) | 103 |
| Region | |
| Northeast | 34 |
| Midwest | 18 |
| South | 35 |
| West | 16 |
| Urban/Rural | |
| Large Central metro | 39 |
| Large fringe metro | 19 |
| Medium metro | 24 |
| Small metro/ Micropolitan/Noncore | 21 |
| Hospital Ownership | |
| Church/Proprietary | 22 |
| Government | 8 |
| Service Type | |
| General acute care | 84 |
| Children's/Psychiatric/Other | 19 |
| Number of Beds | |
| 6-99 | 17 |
| 100-199 | 18 |
| 200-299 | 16 |
| 300-499 | 17 |
| 500-999 | 35 |

**Table 3. Relative importance of top predictive factors across risk domains.**

| Risk Domain | Predictive Factors (% Importance) | | |
| --- | --- | --- | --- |
| | Non-Drug Involved Mortality | Predicting Opioid Involved Mortality | Predicting Progression to Higher Potency, Among Deaths Only |
| Community | 1. Excessive drinking (0.53%)<br>2. Household experiencing housing problems (0.53%)<br>3. Not proficient in English (0.51%) | 1. Violent crime (1.1%)<br>2. Income inequality (1.0%)<br>3. Not proficient in English (0.93%) | 1. Child poverty (1.5%)<br>2. Segregation index (1.4%)<br>3. Violent crime (1.3%) |
| Hospital | 1. Patient origin (1.3%)<br>2. Census division (0.53%)<br>3. Rurality (0.41%) | 1. Census division (1.8%)<br>2. Rurality (0.83%)<br>3. Patient origin (0.81%) | 1. Census division (1.9%)<br>2. Rurality (1.1%)<br>3. Hospital size (1.0%) |
| Individual Drug Related | 1. Morphine encounter (0.97%)<br>2. Hydromorphone encounter (0.61%)<br>3. Fentanyl encounter (0.50%) | 1. Opioid-related disorder ICD (6.2%)<br>2. Poisoning by heroin ICD (2.4%)<br>3. Nicotine-related disorder ICD (1.6%) | 1. Fentanyl encounter (15.5%)<br>2. Morphine encounter (7.3%)<br>3. Hydromorphone encounter (5.5%) |
| Individual Non-drug Related | 1. Neoplastic ICD (22.0%)<br>2. Age (17.9%)<br>3. Cardiovascular ICD (4.0%) | 1. Age (7.3%)<br>2. Primary source of payment (2.2%)<br>3. Secondary source of payment (2.2%) | 1. Age (3.4%)<br>2. Primary source of payment (2.2%)<br>3. Factors influencing health status (2.0%) |

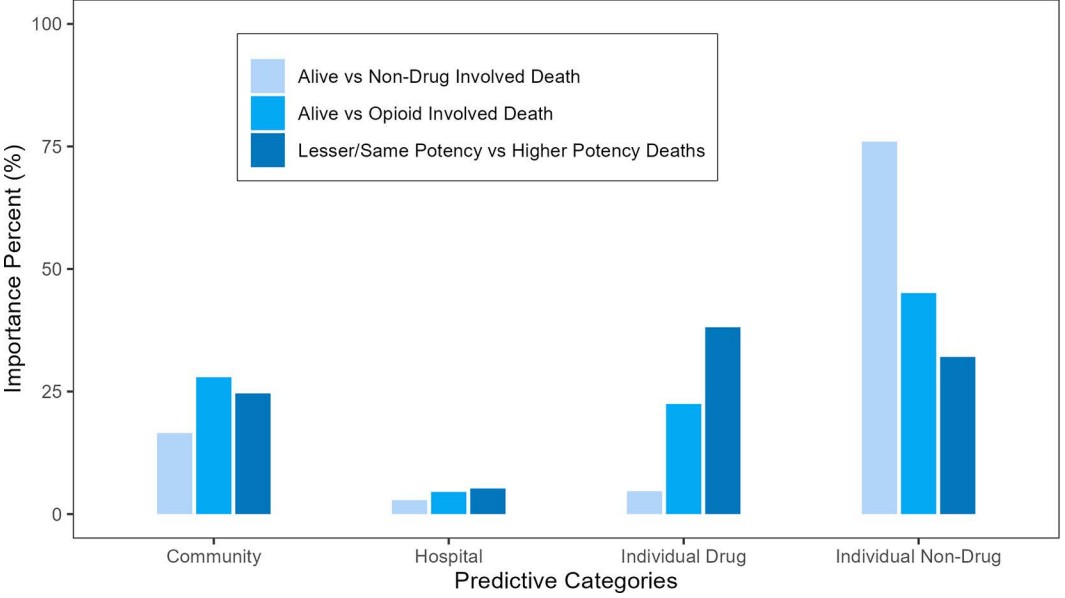

**Fig 1. Importance in predicting opioid-involved mortality by socioecological factor category.**

accounted for only 16.5%; drug related predictors were less than 5% of importance. When predicting progression from a lower potency to a higher potency drug, the individual drug-related factors were the most important grouping (38.1%). Across all models, hospital-related factors accounted for 5% or less of the importance.

## Discussion

This study provides a novel quantitative evaluation of the relative importance of many factors influencing drug-involved mortality, specifically focusing on opioid-related deaths. The findings suggest diverse interventions that target multiple

domains of risk and resilience factors will be needed to combat opioid-involved mortality. While individual drug-related factors remain dominant predictors among non-drug involved deaths, community factors were cumulatively a substantial portion of importance when predicting opioid-involved mortality. Rather than suggesting community factors outweigh individual factors, our results highlight individual vulnerability may be amplified or mitigated by broader environmental factors. Interventions targeting larger social determinants of health may be strongly influential in reducing drug-involved mortality.

Notably, non-drug related factors remained important to all prediction models. Age, for example, accounted for a substantial portion of the importance in predicting drug-related mortality, though much less important than for predicting non-drug-related mortality. This suggests that while drug-related factors are important specific contributors to drug-related mortality, factors related to all-cause mortality may still contribute substantially to risk. Some of these all-cause factors, like age, may be difficult or impossible to intervene upon.

The opioid crisis is as much a societal challenge as it is a personal trajectory. Therefore community-level efforts to reduce incidence of misuse and use disorder, such as increase naloxone availability [20], mental health provider access [21], and drug checking services [22], could produce a substantial reduction in opioid-involved mortality, if they can be made widely accessible and effectively reach individuals in need. Although no specific community factor contributed more than approximately 1% of importance, the combined effect had a similar importance as the combined effect of individual-level factors in predicting opioid-involved mortality. This could suggest that community factors have a cumulative effect, and efforts to broadly improve the general health of communities may reduce drug-involved mortality rates. Additional research is needed to more clearly define domains of influence and should be defined in terms how to target interventions toward different policy sectors. Control for limitations in available data, confounding, and correlations should be examined.

The primary limitation of the study is that data are from 2016–2017, prior to the substantial rise in fentanyl-involved overdose deaths [23]. The presence of fentanyl may strongly increase the importance of the drug-related domain relative to the other domains. Second, only hospital encounters with opioids were included, excluding encounters outside hospitals that may potentially have different influences from community factors. Further research should investigate opioid deaths not encountered in hospitals to gain a more comprehensive understanding of the factors influencing opioid-involved mortality in many social situations. The hospital dataset used was not representative, limiting the generalizability of the findings. Third, there are known challenges associated with using death certificate data [24]. Fourth, the classifier was developed to analyze overall importance of factors within a socioecological model and should not be interpreted in the framework of predictions about individual risk. Fifth, many ecological factors would be better represented by individual factors (e.g., income), which may confound the importance of the community domain with the individual domain. Strong correlations among predictor variables in the same domain grouping may elevate their importance in a random forest model relative to other factors [25]. Future work should explore hierarchical modelling to better represent community vs individual contributions.

## Public health implications

This study demonstrated novel use of a machine learning algorithm to quantify the relative importance of different domains in predicting opioid-involved mortality. While these results provide an initial quantification of how socioecological factors contribute to opioid-involved mortality, they also emphasize the need for a dual approach that targets both community health infrastructure (i.e. access to healthcare, economic stability, and other social determinants) and individual-level health interventions in comprehensive prevention efforts. Most public health challenges in the US arise from interconnected socioecological factors, and understanding the relative contribution of these domains can help guide resource allocation and policy decisions. Future systems-level public health research, such as demonstrated here or through other statistical methodologies [8,26], is critical when integrating real-world policy evaluations and identifying the most effective interventions that improve health outcomes of both of communities individuals.

## Supporting information

**S1 Supporting Methodology.** Additional methodologic specifications.
(DOCX)

## Acknowledgments

The findings and conclusions in this research are those of the authors and do not necessarily represent the views of the Research Data Center, the NCHS, or the Centers for Disease Control and Prevention.

## Author contributions

**Conceptualization:** Joshua C. Black.

**Data curation:** Annika M. Czizik.

**Formal analysis:** Joshua C. Black, Annika M. Czizik.

**Funding acquisition:** Joshua C. Black.

**Validation:** Joshua C. Black.

**Visualization:** Annika M. Czizik.

**Writing – original draft:** Joshua C. Black.

**Writing – review & editing:** Joshua C. Black, Annika M. Czizik.

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
