## [Decision Letter · Decision Letter 0]

PONE-D-24-46562Relative Importance of Socioecological Domains to Predicting Opioid OverdosePLOS ONE

Dear Dr. Black,

Thank you for submitting your manuscript to PLOS ONE. After careful consideration, we invite you to submit a revised version of the manuscript that addresses the points raised during the review process, as we feel that it has merit but does not fully meet PLOS ONE’s publication criteria as it currently stands. **The Reviewers have identified several important methodological limitations and other recommendations that require your attention. I am confident that these changes will strengthen your manuscript, and we look forward to receiving your responses and revisions.**

We look forward to receiving your revised manuscript.

Kind regards,

Andreas Pilarinos, Ph.D., M.P.P., B.Sc.

Academic Editor

PLOS ONE

Journal Requirements:

JCB was supported by the Centers for Disease Control and Prevention through an RFP titled: Call for Proposals for Using Linked Hospital and Mortality Data to address the Opioid Crisis (https://www.cdc.gov/nchs/data/nhcs/FY18-PCORTF-Call-for-Proposals-508.pdf). Funders played no role in the study design, analysis, or decision to publish. The manuscript was reviewed by representatives from the CDC to prevent disclosure of personally identifying information.

This research was conducted by Rocky Mountain Poison and Drug Safety (RMPDS), a division of nonprofit Denver Health and Hospital Authority (DHHA), a political subdivision of the State of Colorado. Outside of this work, RMPDS is supported by subscriptions from pharmaceutical manufacturers, government, and non-government agencies for surveillance, research and reporting services. No subscriber participated in the conception, analysis, drafting, or review of this manuscript.

We note that one or more of the authors are employed by a commercial company. 

“The funder provided support in the form of salaries for authors, but did not have any additional role in the study design, data collection and analysis, decision to publish, or preparation of the manuscript. The specific roles of these authors are articulated in the ‘author contributions’ section.”

4. In the online submission form, you indicated that your data is available only on request from a third party. Please note that your Data Availability Statement is currently missing the contact details for the third party, such as an email address or a link to where data requests can be made. Please update your statement with the missing information. 

Reviewers' comments:

Reviewer's Responses to Questions

**Comments to the Author**

1. Is the manuscript technically sound, and do the data support the conclusions?

Reviewer #1: Yes

Reviewer #2: Partly

2. Has the statistical analysis been performed appropriately and rigorously? 

Reviewer #1: Yes

Reviewer #2: No

3. Have the authors made all data underlying the findings in their manuscript fully available?

Reviewer #1: Yes

Reviewer #2: Yes

4. Is the manuscript presented in an intelligible fashion and written in standard English?

Reviewer #1: Yes

Reviewer #2: Yes

5. Review Comments to the Author

**Reviewer #1:**  Reviewer Comments

This is an important work given the significant role of both socioecological and individual risk factors in influencing opioid-related mortality and I thank the authors for the opportunity to review it. While other studies exist on predicting opioid overdose risk after hospitalization, this is the first to my knowledge that incorporates hospital, individual, and socio-ecological characteristics. It’s an impressive analysis using a variety of data sources.

However, there are a few important points the authors should address. Most significantly, the model’s accuracy seems implausibly high – potentially due to the extremely low rate of deaths in the sample. There should be more information on how the model was evaluated, what data was used for ‘out of bag’ evaluation (i.e., 25% random sample?), any model tuning that was done, and whether the population of interest is only individuals who experienced an opioid-related hospitalization or all 400k+ patients in the sample. Likewise, whether training/testing was done by individual or at the zip code/hospital level is important information given the possibility of spillover between the training and testing samples.

Major Comments:

Abstract:

Major:

You focus on predicting opioid overdose-related mortality. It might make sense to be explicit about this in the title.

While the predictive modeling approach is important to clarify, I think the primary contribution of this manuscript is in the variable importance rankings. I might emphasize that more in the abstract as right now the model performance is centered.

I’m not exactly convinced of the takeaway. Is improving overall community health *really* going to help more than directed targeting? This seems to underplay the importance of individual-level factors seen in the literature.

Introduction:

Minor:

Typo on line 56 – “, though They” – the writing is too colloquial, and the first sentence doesn’t have a period.

Methods:

Major:

How can you say for certain that these ecological factors are not due to individual patients’ own characteristics? There seems to be some confounding in arguing that ecological factors are a primary contributor without the individual’s own health behaviors, economic circumstances, etc. being taken into account.

What does it mean to have these variables in four different domains? Is this conceptual or is this incorporated into the modeling approach somehow?

Could you provide further information on these socioecological models you reference? Why did you choose these and what is the significance of the categories?

Will you do any tuning of the parameters for random forest or are you just using it “out of the box?”

What is the special significance of hospitalizations followed by overdose death caused by more powerful opioid? Why is this a separate category?

How is out of bag performance evaluated? What is the training/testing split in data? Do you split into training/testing at the zip code level or individual patient level? There could be a contagion issue if splitting at individual level due to shared ecological conditions within the same zip code or hospital.

Over what time frame on average are patients followed after the first hospital encounter?

Why focus on non-drug-related mortality? Is this a ‘negative control’ to determine whether the model is producing sensible results for drug-related mortality?

Minor:

Line 111: “The second grouping contained individual drug-related factors were grouped,” – grouping and grouped are redundant here.

Results:

Major:

Is the sample size here really over 400k? If you only focus on individuals who have been hospitalized for opioids previously, isn’t your true sample size 1,144?

How are there only 1,144 individual hospital encounters but over 2k deaths? Aren’t the deaths supposed to be among individuals with a previous hospitalization?

How are you evaluating accuracy? Is this just the number of patients correctly classified/the number of total patients to classify? If so, these values seem implausibly high. One possibility is that overdose deaths are a rare event and so the model could assign everyone to ‘no death’ and look extremely (misleadingly) accurate. For example, if only 1% of individuals die of an opioid overdose, simply classifying everyone as “not dead from opioid overdose” would produce 99% accuracy. It might make sense to choose a better suited metric that accounts for this.

Another reason to doubt this model is the focus on factors that affect drug-related death also resulted in extremely high levels of performance for non-drug mortality. There are certainly some overlapping factors but I’m surprised at this level of performance.

The most important factors identified make sense for both drug- and non-drug-related mortality. However, it’s surprising that non-drug related factors were more important for opioid-related mortality, which seems to run counter to the existing literature. Could you expand on why this might be the case in the discussion?

Discussion:

Major:

I’m not sure I completely buy the conclusion that targeting larger sociodemographic determinants could be *equally* important to influencing individual drug-related behaviors. This seems like overinterpreting what the model can tell you given that it runs counter to the existing literature. I might temper this argument a bit.

You should add the limitation of potential confounding between socioecological factors and individual factors (i.e., economic circumstances of area vs individual).

**Reviewer #2:**  See atatched comments for details.

The term “opioid deaths originating in hospitals” not well-defined nor well-measured in the study:

• People who receive an opioid during a hospital encounter could reflect acute treatment, end of life treatment, or chronic use – all of which do not necessarily lead to death

• We cannot be certain that the death “originated” in the hospital given that the authors define this as individuals “who died in 2016-2017 (primary outcome) during or subsequent to the hospital encounter.” How soon after the hospital encounter is the “subsequent” death? I would restrict the study to deaths occurring in the hospital.

The way “opioid involved death” is interpreted in this study is somewhat misleading given that the motivation of the study is to address the opioid crisis:

• Although the Drug-Involved-Mortality supplements to the National Death Index are relatively reliable for identifying cause-of-death involving opioids, according to the CDC, involvement includes therapeutic use (e.g., end of life) or simply “being on board” which would have a very different interpretation than an overdose. Moreover, the “progression” from lower to high intensity opioid could reflect the standard of care for treatment as the patient nears death (e.g., for cancer) rather than an overdose, creating a mechanical relationship between opioid involved mortality and progression in intensity. I would drop the “progression” outcome from the analysis entirely.

• From https://www.cdc.gov/rdc/b1datatype/datafiles/Drug-Involved-Mortality-Data-Documentation.pdf : “Involvement encompasses everything a substance can do or be used for. This includes overdose (poisoning); adverse reactions; being present on board; therapeutic use; being used, misused, or abused; exposure not otherwise specified; complicating health conditions; interacting with other substances; and causing drug dependence or being used by drug dependent persons. Involvement may occur at or prior to time of death (e.g., “HISTORY OF HEROIN ABUSE”). The underlying assumption is that the substances mentioned on the death certificate contributed to death. However, this assumption may not be true in some cases. For example, a substance mentioned as being present on board may have not contributed to death. In addition, a substance could have been used to treat the health conditions that led to death, but also did not contribute to death. Lastly, the death certificate literal text may not contain sufficient information to ascertain exactly how a substance was involved in death. For example, the literal text may simply mention “HEROIN”, with no contextual information on whether heroin resulted in overdose or whether the decedent had a history of heroin abuse. These cases are also assumed to be substance-involved deaths.

The sample of hospitals in the study does not seem representative.

• The study includes a total of 103 hospitals out of the 7,341 that operate nationwide. Although the study states the sample of hospitals is a mix of rurality, service type, size, ownership, and geographic regions, the question is relative to what? None of the hospitals in the sample are in the Northeast, compared to 14.3% of hospitals nationwide. I would encourage the authors to obtain national statistics on each hospital characteristic to benchmark their sample for the reader rather than just stating it’s not nationally representative.

The results obtained from the population studied do not seem generalizable to understanding the opioid overdose crisis.

• Of the 429,005 patients with opioid-related encounters that were analyzed, more than half (56.6%) were female. This was true with respect to every drug type except heroin, which stands in contrast to CDC data showing that of the 27.9 million ED visits related to drug use in the DAWN database, visits by men were 2.69 times more likely to involve illicit drugs than visits by women (https://pmc.ncbi.nlm.nih.gov/articles/PMC4899061/ ). This suggests to me that the population in this study are patients seeking care for other health-related conditions rather than “fatal opioid overdoses that occurred after hospital encounters involving opioids.”

• Also, while I appreciate that the authors acknowledge that the primary limitation of the study is that data are from 2016-2017, prior to the substantial rise in fentanyl-involved overdose deaths, I do not agree with their assessment that “the relative impact of different factors is potentially unchanged since these data were collected.” Given the swift rise in fentanyl use, misuse and overdose deaths, it is likely or at least plausible that fentanyl now swamps other community factors in terms of relative importance.

The results in Table 3 regarding the relative contribution of the many factors listed are neither novel nor surprising.

• The results in Table 3 do not seem as important as the results summarized in the figure in the supplementary materials. The relative contributions of individual factors are tiny and not all that different from each other in many instances. For example, the top 3 community factors contributing to non-drug involved mortality are: 1. Excessive drinking (0.53%), 2. Household

experiencing housing problems (0.53%), and 3. Not proficient in English (0.51%).

• In other cases, the factors are quite obvious in terms of their importance. For example, in the non-drug involved mortality model, the most important individual factor was neoplastic-related diagnostic codes (e.g. cancer). The top two individual drug-related factors for drug-involved mortality were: 1. Opioid-related disorder ICD (6.2%) and 2. Poisoning by heroin ICD (2.4%). Ditto for the top factors related to progressing to higher potency: 1. Fentanyl encounter (15.5%), 2. Morphine encounter (7.3%), and 3. Hydromorphone encounter (5.5%) I am not sure what is gained by this knowledge as clinicians likely already know that OUD is the top factor leading to drug-involved mortality and that encounters involving fentanyl, morphine, and hydromorphone are likely to lead to deaths involving more potent drugs.

I could not find any table in the paper or supplementary materials that showed this result:

• “Neoplastic diagnostic codes, age, and cardiovascular diagnostic codes accounted for 150 nearly 45% of the importance in model prediction of non-drug related death.”

2. Has the statistical analysis been performed appropriately and rigorously?

One of the major limitations of using a Random Forest ML model is that of correlation bias. Out-Of-Bag prediction accuracy can overestimate the true prediction error when there are a large number of predictor variables or even small correlations between predictors. I would suggest that the study be guided more by the theory of which variables should be included as factors and how they should be measured relative to other factors rather than take the “kitchen sink” approach and include every variable that is available.

• If features are highly correlated, the importance scores might not accurately reflect the true individual contribution of each variable. There are a number of instances where the features in the study are highly correlated:

o The study uses total prescription drug dispensing from the IQVIA Longitudinal Patient Database rather than per capita drug dispensing. Total drug dispensing is likely to be highly correlated with population, meaning there will be a high correlation between the IQVIA variables and the population density variables (e.g., rural population percentage, hospital urban/rural designation).

o Economic variables used in this study (e.g., Unemployment percentage, Children in poverty percentage, Median household income) are all highly correlated with one another as well as other community variables related to socio-economic status (e.g., high school graduation percentage, Violent crime population rate, Households experiencing severe housing problems percentage, Experiencing food insecurity percentage, food environment index).

o The health status variables (e.g., Fair or poor health percentage, Number of poor physical health days, Number of poor mental health days, Frequent physical distress percentage, Frequent mental distress percentage, Preventable hospital stay rate) are all highly correlated with each other and with the health behavior variables (e.g., Adult smoking percentage, Physically inactive percentage, Access to exercise opportunity percentage, Excessive drinking percentage).

The study also suffers from omitted variable bias, particularly the lack of demographic characteristics and health care delivery system variables that have been shown to affect opioid prescribing. I would encourage the authors to include these county level variables that are available from the Area Health Resource File:

• While the study includes age and gender, there are large disparities in opioid prescribing and overdose deaths across communities based on the non-white population rate and the veteran population rate.

• Similarly, while the study includes the number of providers for certain specialties (e.g., Primary care physician population rate, Mental health providers population rate), it fails to capture other important factors such as the availability of skilled nursing facilities per population and hospital beds per population, that can affect the likelihood of access to treatment.

Overall, while the Random Forest method may seem appealing in working with such a large dataset with many factors, it is also limiting in its ability to tell us anything about the magnitude of the effect. I would encourage the authors to use an Oaxaca decomposition to be able to capture both the absolute magnitude and relative contribution of each factor in predicting mortality.

• Otherwise, I worry that the study leads to false conclusions such as this, where the authors assume that community factors are more important but might be picking up on the many inter-correlated factors in that particular grouping: “Although no specific community factor contributed more than approximately 1% of importance, the combined effect was more important than individual-level factors in predicting opioid-involved overdose. This could suggest that community factors have a cumulative effect, and efforts to broadly improve the general health of communities may reduce drug overdose rates as much as drug-specific interventions.”

3. Have the authors made all data underlying the findings in their manuscript fully available? The PLOS Data policy requires authors to make all data underlying the findings described in their manuscript fully available without restriction, with rare exception (please refer to the Data Availability Statement in the manuscript PDF file). The data should be provided as part of the manuscript or its supporting information, or deposited to a public repository. For example, in addition to summary statistics, the data points behind means, medians and variance measures should be available. If there are restrictions on publicly sharing data—e.g. participant privacy or use of data from a third party—those must be specified.

Restricted data are available through the CDC Research Data Center (RDC). Access

can be obtained through submission of proposals to the RDC.

4. Is the manuscript presented in an intelligible fashion and written in standard English?

Specific errors:

Line 55: missing “and”

“sociodemographic, and past drug use history from among a wide variety of potential predictors (9-11) as”

Line 56: no capitalization of “They”

“important personal factors, though they have been limited to primarily including only personal domain”

6. PLOS authors have the option to publish the peer review history of their article (what does this mean? ). If published, this will include your full peer review and any attached files.

**Do you want your identity to be public for this peer review?** For information about this choice, including consent withdrawal, please see our Privacy Policy .

Reviewer #1: **Yes: ** Robert C. Schell

Reviewer #2: No

---

## [Author Response · Author response to Decision Letter 1]

7 Mar 2025

Reviewer #1 Comments:

Comment 1: You focus on predicting opioid overdose-related mortality. It might make sense to be explicit about this in the title.

Author Response: We have updated the title.

Comment 2: While the predictive modeling approach is important to clarify, I think the primary contribution of this manuscript is in the variable importance rankings. I might emphasize that more in the abstract as right now the model performance is centered.

Author Response: Thank you for the suggestion. We have emphasized the importance more in the abstract.

Comment 3: I’m not exactly convinced of the takeaway. Is improving overall community health *really* going to help more than directed targeting? This seems to underplay the importance of individual-level factors seen in the literature.

Author Response: We don’t want to overemphasize community factors at the expense of individual factors. We have softened the conclusions to make this point. We have also emphasized the potential heightened individual vulnerability that may be amplified by community factors.

Comment 4: Typo on line 56 – “, though They” – the writing is too colloquial, and the first sentence doesn’t have a period.

Author Response: We have updated this sentence.

Comment 5: How can you say for certain that these ecological factors are not due to individual patients’ own characteristics? There seems to be some confounding in arguing that ecological factors are a primary contributor without the individual’s own health behaviors, economic circumstances, etc. being taken into account.

Author Response: All of the ecological factors are aggregates of community measures, so the influence of any individual on the overall aggregate is small and negligible. However, we do note the criticism that ecologically, the reverse is also true that the aggregate may have a small influence on an individual (e.g., the median income has little effect on the individual’s income). We have expanded our limitation on this.

Comment 6: What does it mean to have these variables in four different domains? Is this conceptual or is this incorporated into the modeling approach somehow?

Author Response: This is a conceptual framework developed by Jalali (DOI: 10.1186/s12961-020-00598-6). We have clarified this is not part of the statistical model in the methods.

Comment 7: Could you provide further information on these socioecological models you reference? Why did you choose these and what is the significance of the categories?.

Author Response: We have added detail to the methods. These frameworks represent different levels of intervention where policy may be targeted.

Comment 8: Will you do any tuning of the parameters for random forest or are you just using it “out of the box?”

Author Response: Because the model performed well “out of the box,” we did not do additional tuning of the classifier. We have also added additional detail on the model parameters (e.g., out-of-bag percentages).

Comment 9: What is the special significance of hospitalizations followed by overdose death caused by more powerful opioid? Why is this a separate category?

Author Response: We wished to identify potential differences in importance across domains for mortality related to any opioid death as compared to mortality that resulted from progression to more powerful opioids. We have added clarity to the objective section in the introduction.

Comment 10: How is out of bag performance evaluated? What is the training/testing split in data? Do you split into training/testing at the zip code level or individual patient level? There could be a contagion issue if splitting at individual level due to shared ecological conditions within the same zip code or hospital.

Author Response: Training data was split at the individual level. We have noted in the future work section that hierarchical models may better represent the community-level contribution.

Comment 11: Over what time frame on average are patients followed after the first hospital encounter?

Author Response: The CDC linked dataset (NCHS-NDI-DIM) is a 2-year follow-up from exposure in 2016 to death occurring in 2016-2017. This is noted in the methods.

Comment 12: Why focus on non-drug-related mortality? Is this a ‘negative control’ to determine whether the model is producing sensible results for drug-related mortality?

Author Response: Yes, this acts as a negative comparator in the study. We have clarified this in the methods.

Comment 13: Line 111: “The second grouping contained individual drug-related factors were grouped,” – grouping and grouped are redundant here.

Author Response: We have updated the sentence.

Comment 14: Is the sample size here really over 400k? If you only focus on individuals who have been hospitalized for opioids previously, isn’t your true sample size 1,144?

Author Response: Yes, this was the total sample originally investigated for predicting death. The n=429,005 represents the total exposed population, which is the population at risk. The classifier determines factors predicting death vs not death, and therefore was trained on the entire 400k+ sample.

Comment 15: How are there only 1,144 individual hospital encounters but over 2k deaths? Aren’t the deaths supposed to be among individuals with a previous hospitalization?

Author Response: The n=1,144 represents total patients with specific opioids listed in both the hospital records and the death records. In both of those data sources, a generic code (e.g., “opioids”) may be used. That sample size represents the pool of data with which to assess progression in potency, and missing data may bias those values.

Comments 16 & 17: How are you evaluating accuracy? Is this just the number of patients correctly classified/the number of total patients to classify? If so, these values seem implausibly high. One possibility is that overdose deaths are a rare event and so the model could assign everyone to ‘no death’ and look extremely (misleadingly) accurate. For example, if only 1% of individuals die of an opioid overdose, simply classifying everyone as “not dead from opioid overdose” would produce 99% accuracy. It might make sense to choose a better suited metric that accounts for this.

Another reason to doubt this model is the focus on factors that affect drug-related death also resulted in extremely high levels of performance for non-drug mortality. There are certainly some overlapping factors but I’m surprised at this level of performance

Author Response: We agree with the reviewer’s concern. Accuracy was evaluated using an average prediction across the out-of-bag samples in the whole sample, and therefore does represent a simple assessment of accuracy. Because our goal here was to evaluate importance in prediction only, rather than make predictions about an individual’s likelihood of death, we believe the out-of-bag prediction accuracy is sufficient to describe the model performance. We have added commentary in the discussion to ensure readers to not misconstrue the predictive accuracy of the classifier.

Comment 18: The most important factors identified make sense for both drug- and non-drug-related mortality. However, it’s surprising that non-drug related factors were more important for opioid-related mortality, which seems to run counter to the existing literature. Could you expand on why this might be the case in the discussion?

Author Response: Thank you for the suggestion. We have expanded on this in the discussion.

Comment 19: I’m not sure I completely buy the conclusion that targeting larger sociodemographic determinants could be *equally* important to influencing individual drug-related behaviors. This seems like overinterpreting what the model can tell you given that it runs counter to the existing literature. I might temper this argument a bit.

Author Response: We agree with the criticism, particularly when comparing total percent importances across models. We have tempered this argument as suggested.

Comment 20: You should add the limitation of potential confounding between socioecological factors and individual factors (i.e., economic circumstances of area vs individual).

Author Response: We thank the reviewer for pointing this out and have added this as a limitation.

Reviewer #2 Comments:

Comment 1: The term “opioid deaths originating in hospitals” not well-defined nor well-measured in the study:

• People who receive an opioid during a hospital encounter could reflect acute treatment, end of life treatment, or chronic use – all of which do not necessarily lead to death

• We cannot be certain that the death “originated” in the hospital given that the authors define this as individuals “who died in 2016-2017 (primary outcome) during or subsequent to the hospital encounter.” How soon after the hospital encounter is the “subsequent” death? I would restrict the study to deaths occurring in the hospital.

Author Response: We agree with the reviewer that the term “originating” is misleading and we have adjusted the language in the manuscript to be more simply “encountering” an opioid.

Comment 2: The way “opioid involved death” is interpreted in this study is somewhat misleading given that the motivation of the study is to address the opioid crisis:

• Although the Drug-Involved-Mortality supplements to the National Death Index are relatively reliable for identifying cause-of-death involving opioids, according to the CDC, involvement includes therapeutic use (e.g., end of life) or simply “being on board” which would have a very different interpretation than an overdose. Moreover, the “progression” from lower to high intensity opioid could reflect the standard of care for treatment as the patient nears death (e.g., for cancer) rather than an overdose, creating a mechanical relationship between opioid involved mortality and progression in intensity. I would drop the “progression” outcome from the analysis entirely.

• From : “Involvement encompasses everything a substance can do or be used for. This includes overdose (poisoning); adverse reactions; being present on board; therapeutic use; being used, misused, or abused; exposure not otherwise specified; complicating health conditions; interacting with other substances; and causing drug dependence or being used by drug dependent persons. Involvement may occur at or prior to time of death (e.g., “HISTORY OF HEROIN ABUSE”). The underlying assumption is that the substances mentioned on the death certificate contributed to death. However, this assumption may not be true in some cases. For example, a substance mentioned as being present on board may have not contributed to death. In addition, a substance could have been used to treat the health conditions that led to death, but also did not contribute to death. Lastly, the death certificate literal text may not contain sufficient information to ascertain exactly how a substance was involved in death. For example, the literal text may simply mention “HEROIN”, with no contextual information on whether heroin resulted in overdose or whether the decedent had a history of heroin abuse. These cases are also assumed to be substance-involved deaths.

Author Response: We agree with the reviewer regarding terminology is overstated based on the DIM data and have rewritten the article to replace “overdose” with “opioid-involved”. We retained the term “overdose” only in cases where it was used to cite another study that did examine overdose.

We do agree that the DIM data is not a nuanced assessment of causal involvement of an opioid, and we have expanded the methods to clarify this assumption from the DIM documentation the reviewer quotes. However, this assumption within the DIM data is not without justification, as the drugs are listed in the cause-of-death fields, which are to be used by the medica certifier for substances suspected to contribute to the death, as described in the DIM development publication (https://pubmed.ncbi.nlm.nih.gov/27996933/). Limitations in using death certification data in a broad sense are cited in the limitations section.

Comment 3: The sample of hospitals in the study does not seem representative.

• The study includes a total of 103 hospitals out of the 7,341 that operate nationwide. Although the study states the sample of hospitals is a mix of rurality, service type, size, ownership, and geographic regions, the question is relative to what? None of the hospitals in the sample are in the Northeast, compared to 14.3% of hospitals nationwide. I would encourage the authors to obtain national statistics on each hospital characteristic to benchmark their sample for the reader rather than just stating it’s not nationally representative.

Author Response: The label “Northwest” was a typo in the manuscript and has been corrected to “Northeast”. Our description of the data source as simply not representative is consistent with reference 13 from NCHS (). The mix of factors are described in Table 2 and describe simply that the sample was diverse based on various characteristics. It is relative only to itself.

Comment 4: The results obtained from the population studied do not seem generalizable to understanding the opioid overdose crisis.

• Of the 429,005 patients with opioid-related encounters that were analyzed, more than half (56.6%) were female. This was true with respect to every drug type except heroin, which stands in contrast to CDC data showing that of the 27.9 million ED visits related to drug use in the DAWN database, visits by men were 2.69 times more likely to involve illicit drugs than visits by women (https://pmc.ncbi.nlm.nih.gov/articles/PMC4899061/ ). This suggests to me that the population in this study are patients seeking care for other health-related conditions rather than “fatal opioid overdoses that occurred after hospital encounters involving opioids.”

• Also, while I appreciate that the authors acknowledge that the primary limitation of the study is that data are from 2016-2017, prior to the substantial rise in fentanyl-involved overdose deaths, I do not agree with their assessment that “the relative impact of different factors is potentially unchanged since these data were collected.” Given the swift rise in fentanyl use, misuse and overdose deaths, it is likely or at least plausible that fentanyl now swamps other community factors in terms of relative importance.

Author Response: We agree that the dataset is not representative and therefore the findings have limited generalizability. We have expanded on this limitation in the discussion, including adjusting our description of the impact fentanyl may have on the relative contributions.

Comment 5: The results in Table 3 regarding the relative contribution of the many factors listed are neither novel nor surprising.

• The results in Table 3 do not seem as important as the results summarized in the figure in the supplementary materials. The relative contributions of individual factors are tiny and not all that different from each other in many instances. For example, the top 3 community factors contributing to non-drug involved mortality are: 1. Excessive drinking (0.53%), 2. Household

experiencing housing problems (0.53%), and 3. Not proficient in English (0.51%).

• In other cases, the factors are quite obvious in terms of their importance. For example, in the non-drug involved mortality model, the most important individual factor was neoplastic-related diagnostic codes (e.g. cancer). The top two individual drug-related factors for drug-involved mortality were: 1. Opioid-related disorder ICD (6.2%) and 2. Poisoning by heroin ICD (2.4%). Ditto for the top factors related to progressing to higher potency: 1. Fentanyl encounter (15.5%), 2. Morphine encounter (7.3%), and 3. Hydromorphone encounter (5.5%) I am not sure what is gained by this knowledge as clinicians likely already know that OUD is the top factor leading to drug-involved mortality and that encounters involving fentanyl, morphine, and hydromorphone are likely to lead to deaths involving more potent drugs.

Author Response: We agree that the individual factors themselves are less novel relative to other literature. We detail much of this prior literature in the introduction. However, our contribution of qu

---

## [Decision Letter · Decision Letter 1]

Relative importance of socioecological domains to predicting opioid-involved mortality

PONE-D-24-46562R1

Dear Dr. Black

We’re pleased to inform you that your manuscript has been judged scientifically suitable for publication and will be formally accepted for publication once it meets all outstanding technical requirements.

Kind regards,

Andreas Pilarinos, Ph.D., M.P.P., B.Sc.

Academic Editor

PLOS ONE

Additional Editor Comments (optional):

Thank you for addressing the Reviewers' comments. We believe your revisions have strengthened the manuscript and look forward to advancing your manuscript to publication.

Reviewers' comments:

Reviewer's Responses to Questions

**Comments to the Author**

1. If the authors have adequately addressed your comments raised in a previous round of review and you feel that this manuscript is now acceptable for publication, you may indicate that here to bypass the “Comments to the Author” section, enter your conflict of interest statement in the “Confidential to Editor” section, and submit your "Accept" recommendation.

Reviewer #1: All comments have been addressed

2. Is the manuscript technically sound, and do the data support the conclusions?

Reviewer #1: Yes

3. Has the statistical analysis been performed appropriately and rigorously? 

Reviewer #1: Yes

4. Have the authors made all data underlying the findings in their manuscript fully available?

Reviewer #1: No

5. Is the manuscript presented in an intelligible fashion and written in standard English?

Reviewer #1: Yes

6. Review Comments to the Author

Reviewer #1: The authors have done a wonderful job addressing my comments and I think this manuscript is ready to publish. Great work!

7. PLOS authors have the option to publish the peer review history of their article (what does this mean? ). If published, this will include your full peer review and any attached files.

**Do you want your identity to be public for this peer review?** For information about this choice, including consent withdrawal, please see our Privacy Policy .

Reviewer #1: **Yes: ** Robert C. Schell

---

## [Editor Report · Acceptance letter]

PONE-D-24-46562R1

PLOS ONE

Dear Dr. Black,

I'm pleased to inform you that your manuscript has been deemed suitable for publication in PLOS ONE. Congratulations! Your manuscript is now being handed over to our production team.

Kind regards,

on behalf of

Dr. Kimberly Page

Academic Editor

PLOS ONE